# A 2D Waveguide Method for Lithography Simulation of Thick SU-8 Photoresist

**DOI:** 10.3390/mi11110972

**Published:** 2020-10-29

**Authors:** Zi-Chen Geng, Zai-Fa Zhou, Hui Dai, Qing-An Huang

**Affiliations:** Key Laboratory of MEMS of the Ministry of Education, Southeast University, Nanjing 210096, China; 220184753@seu.edu.cn (Z.-C.G.); 220184865@seu.edu.cn (H.D.); hqa@seu.edu.cn (Q.-A.H.)

**Keywords:** lithography simulation, microelectromechanical system, waveguide method

## Abstract

Due to the increasing complexity of microelectromechanical system (MEMS) devices, the accuracy and precision of two-dimensional microstructures of SU-8 negative thick photoresist have drawn more attention with the rapid development of UV lithography technology. This paper presents a high-precision lithography simulation model for thick SU-8 photoresist based on waveguide method to calculate light intensity in the photoresist and predict the profiles of developed SU-8 structures in two dimension. This method is based on rigorous electromagnetic field theory. The parameters that have significant influence on profile quality were studied. Using this model, the light intensity distribution was calculated, and the final resist morphology corresponding to the simulation results was examined. A series of simulations and experiments were conducted to verify the validity of the model. The simulation results were found to be in good agreement with the experimental results, and the simulation system demonstrated high accuracy and efficiency, with complex cases being efficiently handled.

## 1. Introduction

### 1.1. Research Background

The lithography process refers to the process of transferring geometric pattern on the mask to the photoresist coated on the surface of the semiconductor wafer. With the rapid development of semiconductor technology, the lithography process has become an important propulsive force to achieve integrated and miniaturized devices. SU-8 photoresist [1] is a chemically amplified negative-tone resist based on epoxy resin, which has excellent chemical stability, resistance against solvents, and mechanical and optical properties [2]. It has become a mainstream micromachining technology for high aspect ratio structures in the field of microelectromechanical systems (MEMS) [3], including micrototal analysis systems [4], microfluidic systems [5], electroplating [6], and so on.

The propagation process of light during exposure is shown in Figure 1. First, we assume that the incident light is parallel, uniform, and perpendicular to the mask surface. The incident light is diffracted when passing through the mask hole. Then, when the light reaches the interface between the air and the photoresist, part of the light will be refracted and will continue to propagate through the photoresist, and the other part will be reflected back into the air. Part of the light that continues to propagate to the inside of the photoresist will be absorbed by the photoresist and generate photoacid at the same time. Finally, at the photoresist/substrate interface, part of the light is refracted into the substrate, and part of the light is reflected back into the photoresist and superimposed with the incident light. In general, the entire propagation process can be regarded as the propagation of diffracted light, taking into account other optical phenomena. Among them, the diffraction effect has the most important influence on light intensity distribution in the photoresist.

### 1.2. Research Significance

Because of expensive equipment and complicated process steps in the lithography process, we need repeated experiments, which is costly and time consuming. It is difficult to understand the inherent principle of lithography technology. Therefore, lithography simulation technology [7] has become a useful tool for predicting and optimizing the manufacturing process, which is also an integral part of lithography technology. Lithography simulation technology avoids the problems of high cost and time consumption caused by repeated plate making and tape production. At the same time, it can optimize the process parameters and promote the development of lithography technology. The flow chart of lithography simulation is as follows (Figure 2).

The final morphology of thick SU-8 photoresist after development depends largely on the exposure process [8]. The developed photoresist morphology can be predicted by simulating light intensity distribution in the photoresist. Therefore, it has become an inevitable and useful requirement to establish a lithography simulation model [9] of SU-8 thick photoresist to calculate light intensity distribution.

In this paper, a simulation model for UV lithography of SU-8 negative thick photoresist is presented to predict the profiles of developed SU-8 structures based on rigorous electromagnetic field theory. The model schematic is shown in Figure 3. We used the light intensity distribution calculation model to calculate the distribution of light intensity based on mask information, light source information, and photoresist information. The parameters of the air gap, incident angle, and photoresist depth, which have significant influence on the profile quality of developed structures, were studied in the simulation. Simulation results based on profiles of exposure intensity distribution were compared with the experimental results to validate the model. 

## 2. Materials and Methods

Light intensity simulations are based on two main categories of models. One category is based on scalar diffractive theory [10], and the other is based on rigorous electromagnetic field theory [11], including the finite difference time domain method (FDTD) and the waveguide (WG) method. The limitation of these models comes from simulation accuracy, application capability for special lithography methods such as inclined SU-8 photoresist lithography, and numerical costs for large-scale 3D simulations. Therefore, considering calculation accuracy and calculation time, we finally used the waveguide method to calculate light intensity distribution.

The waveguide [12,13] method has been developed and employed in lithography simulations during the past decade. The idea of the waveguide method is based on rigorous electromagnetic field theory, and it is well adapted to lithography simulation with many typical mask geometries. The first prototype of the waveguide method was presented by Nyyssonen [14] in order to simulate the measurement of optical linewidth. Then, this method was used to simulate diffraction when light passes through a 2D-phase shift mask based on Yuan’s theory [15] and later extended to 3D domain [16] by Lucas.

The model is based on the 2D spatial frequency solution to Maxwell’s equations, the so-called waveguide method [17,18,19,20,21]. Firstly, the simulated structure is sliced into different rectangular layers with homogeneous optical properties in the *Z*-axis direction, as shown in Figure 4. Then, the material parameters and electromagnetic fields in each layer are expanded by Fourier series. Simultaneous equations formed by these expanded series and Maxwell equations give an eigenvalue problem. After that, a hybrid approach using vector potentials is applied to couple electromagnetic field in each layer. Finally, each layer is coupled according to the proper boundary continuity condition, and the electromagnetic field information in the photoresist is converted into light intensity information. 

Firstly, the coupled Maxwell’s equation determined by vector potential [22] can be simplified to two differential equations by separating the variable; the equation contains the complex potential $k^{2}$ε and the eigenvalue $\alpha^{2}$:(1)$$\begin{array}{l} \frac{d^{2}X}{dx^{2}}+k^{2}\varepsilon X=\alpha^{2}X\\ \frac{d^{2}Z}{dz^{2}}=-\alpha^{2}Z \end{array}$$

Before solving the equation numerically, the dielectric constant ε needs to be expanded by Fourier series. It is assumed that the mask structure is arranged infinitely and repeatedly, with the length *d* as the period in the x-direction:(2)$$\varepsilon^{j}(x)=\sum_{q=-L}^{+L}\varepsilon_{q}^{j}\exp(i2\pi qbx)$$

The coefficient of each term can be obtained by inverse Fourier transform:
(3)$$\varepsilon_{q}^{j}=b\int_{0}^{d}\varepsilon^{j}(x)\exp(-i2\pi qbx)dx\;X\left(x\right)=\underset{l}{\sum }B_{l}\exp\left(i2\pi lx/d\right)$$


Then, the material parameters and electromagnetic fields in each layer are expanded by Fourier series. Simultaneous equations formed by these expanded series and Maxwell equations give an eigenvalue problem:(4)$$\begin{array}{l} DB=\alpha^{2}B\\ D_{pq}=k^{2}\varepsilon_{p-q}-{\left(2\pi p/d\right)}^{2}\delta_{p,q} \end{array}$$
where *α* and *B* are the eigenvalues and eigenvectors of a (2L + 1) by (2L + 1) matrix *D*. The computation time depends on the number of orders of the Fourier series expansion and the number of rectangular layers.

The electromagnetic field distribution in layer *j* is as follows. $\alpha_{m}^{j},B_{m}^{j}$ is obtained by solving the eigenvalue problem, and $A_{m}^{j},A^{\prime }{}_{m}^{j}$ is obtained using the boundary continuity condition: (5)$$\left\{\begin{array}{l} \;E_{y}^{i}=\sum_{m}\left[A_{m}^{j}\exp(i\;\alpha_{m}^{j}(z-z_{j}))+A_{m}^{\prime j}\exp(-i\;\alpha_{m}^{j}(z-z_{j}))\right]\times \sum_{l}B_{l,m}^{j}\exp(i2\pi lx/d)\\ \;H_{x}^{j}=\frac{i}{k}\frac{\partial E_{y}^{i}}{\partial z}\;H_{z}^{j}=-\frac{i}{k}\frac{\partial E_{y}^{i}}{\partial x} \end{array}\right.$$

After making this transformation and solving the eigenvalue problem, the boundary continuity conditions are matched so that the tangential components of the electric and magnetic fields have to be continuous at *z* = 0 [23]. The boundary continuity condition is as follows:(6)$$E_{y}^{j}=E_{y}^{j+1}\;H_{y}^{j}=H_{y}^{j+1}$$

In the air and first layer interface:(7)$$\left[\begin{array}{l} C_{11}^{0}{\;C}_{12}^{0}\\ \;0\;0 \end{array}\right]\left[\begin{array}{l} A^{1}\\ A^{\prime }{}^{1} \end{array}\right]=\left[\begin{array}{l} R\\ 0 \end{array}\right]$$(8)$$\begin{array}{ll} R_{l} & =2{\left[1-{\left(l_{0}b\lambda_{0}\right)}^{2}\right]}^{\frac{1}{2}}\delta_{l,l_{0}}\\ & =2{\left[1-{\left(\sin\theta \right)}^{2}\right]}^{\frac{1}{2}}\delta_{l,l_{0}} \end{array}$$
where θ is the incident angle, and $l_{0}$ is the order of incident plane wave. It is convenient for us to change the different parameters to obtain the corresponding results.

By matching the boundary conditions at *z* = *T*, we obtain the following:(9)$$\left[\begin{array}{l} C_{11}^{n}{\;C}_{12}^{n}\\ \;0\;0 \end{array}\right]\left[\begin{array}{l} A^{n}\\ A^{\prime }{}^{n} \end{array}\right]=\left[\begin{array}{l} 0\\ 0 \end{array}\right]$$

At an intermediate interface *z* = *z_j_*, the waves in layer *j*+1 are related to the waves in layer *j* by
(10)$$\left[\begin{array}{l} C_{11}^{j}{\;C}_{12}^{j}\\ C_{21}^{j}{\;C}_{22}^{j} \end{array}\right]\left[\begin{array}{l} A^{j}\\ A^{\prime }{}^{j} \end{array}\right]=\left[\begin{array}{l} E_{11}^{j+1}{\;E}_{12}^{j+1}\\ E_{21}^{j+1}{\;E}_{22}^{j+1} \end{array}\right]\left[\begin{array}{l} A^{j+1}\\ A^{\prime }{}^{j+1} \end{array}\right]$$
where matrix *C* contains only the topography information, such as mask size, material parameters, and so on, and the vector [*A*, *A′*] contains diffraction information.

Finally, the light intensity can be calculated by the electric field value:(11)$$I(p,q)=n_{r}{\left|E_{y}^{j}(p,q)\right|}^{2}$$

The incident angle is due to refraction at the air/photoresist interface. The structural inclined angle and the structural width of the developed photoresist vary with the incident angle. The relationship between the incident angle *δ*, the refractive angle *θ*, and the structural inclined angle *α* is determined by Snell’s law as follows:(12)$$\Theta =\sin^{-1}\left(n_{1}\;\cdot \;\sin\delta /n_{2}\right)$$
(13)α=90−θ
where *n**_1_* (= 1) and *n**_2_* (= 1.67) are the refractive indices of the air and SU-8, respectively.

More details of the method is described in reference [24,25,26].

## 3. Results and Discussion

### 3.1. Simulation Results

The simulation results showed the distribution of normalized intensity calculated under different conditions. In addition to finding light intensity distribution on a certain layer of photoresist, a contour map can also describe the intensity value of a two-dimensional profile of photoresist. This method can help intuitively predict the final development morphology of the photoresist. Observing the influence of different parameters on light intensity distribution provides a guide to set the corresponding process parameters so as to optimize the structure design.

In the exposure process, the surface photolithographic dose *D* (mJ/cm^2^) is multiplied by the exposure time and the incident light intensity. The surface exposure dose can be considered to be constant over the entire thickness of the photoresist layer. Photolithography of thick-film photoresist implies that the thickness of the photoresist material needs to be taken into account in order to determine the optimal exposure dose for a given photoresist film thickness. The exposure dose is the light intensity at a given thickness and exposure time. According to the material parameters of the photoresist and substrate, the depth of the photoresist can be calculated to obtain the optimal exposure dose under the given photoresist depth. In this calculation, the incident light intensity was 2.6 mW/cm^2^, its wavelength was 365 nm, and the thickness of SU-8 photoresist was 300 μm. According to our simulation conditions, when the thickness of SU-8 photoresist is 300 μm, the optimal exposure dose is 140 (mJ/cm^2^) [27].

First, we analyzed the simulation of the intensity of light at vertical incidence. The intensity curves for vertical exposure with different thicknesses are shown in Figure 5, and the corresponding contour maps are shown in Figure 6. We can see from the results that the intensity in SU-8 was simultaneously attenuated along the radiation direction with increasing photoresist thickness because of the diffraction and absorption of the incident light in the photoresist.

The previous simulation was based on the case of a single mask hole. In practice, there are often multiple mask holes, and selection of the spacing between the mask holes is an important parameter to be investigated when designing the layout. Different mask hole spacing will affect the final development morphology of the photoresist. Therefore, Figure 7 shows the intensity distribution of the photoresist in the case of multiple mask holes, and the corresponding contour maps are shown in Figure 8. The mask hole spacing was 40 μm, and the mask hole size was 70 μm. The length of the photoresist was 200 μm, and the abscissa in the figure represents the horizontal position. According to the proportion calculation, it can be seen that the simulation results are correct.

The above simulation results were obtained without considering the substrate reflection. Part of the light is reflected at the photoresist and substrate interfaces. The refractive index can be calculated according to the refractive index of the photoresist and the substrate, thereby enabling calculation of the distribution of light intensity after refraction. During vertical UV lithography processes of SU-8, reflection enhances light intensity at the same position as the photoresist. During inclined UV lithography processes of SU-8, the incident UV light is reflected at the SU-8/substrate interface. The reflected UV light will be transmitted into other unexposed SU-8, usually creating reflected induced structures. The light intensity distribution under different conditions at the same position is shown in Figure 9 and Figure 10. By comparing the value of light intensity with and without reflection at the same position, we found that the light intensity considering substrate reflection was greater than that without substrate reflection, and this phenomenon was more obvious at the bottom of the photoresist. This is consistent with the previous analysis. 

One of the advantages of substrate reflection is that it can ensure the photoresist bottom is fully exposed, avoiding the disadvantage of the photoresist bottom not being fully exposed due to it being too thick. It should be noted that if the substrate reflection is not well controlled, it will lead to overexposure of the photoresist bottom, which in turn will lead to deviation between the results and the design. The reflection at the photoresist/substrate interface cannot be neglected as the dose of reflected UV light is increased by increasing the exposure time to more than that of a regular exposure process. To eliminate the reflected induced patterns, we can coat a layer of antireflective film on the photoresist or substrate surface, called TARC and BARC, respectively.

Secondly, we analyzed the simulation of light intensity at oblique incidence. The incident UV light was inclined from left to right, and the incident light intensity was 2.6 mW/cm^2^. Other conditions were the same as vertical incidence.

The intensity curves for inclined exposure with different air gaps are shown in Figure 11, and the corresponding contour maps are shown in Figure 12. As can be seen, as the air gap increased from 10 to 30 μm, the exposure area shifted to the right, the diffraction effect became more intense, and the edge diffraction effect also became severe. 

Figure 13 illustrates the intensity curves resulting from different incident angles, and the corresponding contour maps are shown in Figure 14. We can see from the results that the exposure area shifted to the right, and the light intensity began to decrease with increasing inclined angle. Analyzing this phenomenon from a mathematical point of view, according to the expression of matrix *R* in Formula (8), the value of *R* decreased with the increase in incident angle, and therefore the final light intensity also decreased. 

The intensity curves for inclined exposure with different thicknesses are shown in Figure 15, and the corresponding contour maps are shown in Figure 16. It can be seen from the Figure 15 that due to diffraction of light and absorption of the photoresist, the light intensity was different at different depths. On the surface of the photoresist, the light intensity was the highest, while it was the smallest on the bottom of the photoresist. It should be noted that the intensity in SU-8 was simultaneously attenuated along the radiation direction with increasing photoresist thickness. 

Compared with vertical incidence, the intensity curve under oblique incidence decreased from top to bottom and shifted to the right. Further simulation showed that this translation amplitude would be larger with the increase of incident angle. Finally, the computation time depended on the number of orders of the Fourier series expansion. With an increase of the expansion order, the calculation accuracy was higher and the calculation time increased accordingly. Therefore, in actual simulation, it is necessary to select the appropriate expansion order in order to consider the calculation time and calculation accuracy.

### 3.2. Experimental Results and Analysis

Experiments for UV lithography were conducted to validate the proposed lithography simulation model, and simulation results based on exposure intensity distribution patterns are illustrated for demonstration. In this calculation, the incident light intensity was 2.6 mW/cm^2^, and its wavelength was 365 nm. The thickness of SU-8 photoresist was 300 μm, and the thickness of the air gap was 10 μm. Besides, in order to efficiently evaluate and compare the simulation and experimental results, the contour value of light intensity patterns was assumed as 0.38 in this study.

SEM photos of SU-8 structures fabricated on glass wafers are illustrated in Figure 17a, and the corresponding simulation results are shown in Figure 17b. From Figure 17a, we can see that the width of the top SU-8 was 31.37 μm, and the width of the bottom SU-8 was 32.06 μm. Therefore, after development, the width of SU-8 pillars was enlarged over the entire height of the photoresist. In particular, the variation of the bottom width was 0.76 μm. Due to the absorption of light by photoresist, for a relatively low exposure dose near the bottom of the SU-8 photoresist layer, the width at the bottom is slightly larger than that at the top. By means of threshold prediction (the contour value was assumed as 0.38) of the light intensity pattern, the top and bottom widths derived from the simulation result in Figure 17b were 33.28 and 34.7 μm, respectively. In comparison, the top width in the intensity pattern was approximately equal to that in the experimental result. Therefore, we drew the conclusion that the simulation results were in good agreement with the experimental results.

Figure 18 and Figure 19 show a series of SU-8 pillars with different proportions of linewidth and spacewidth and the corresponding simulation results of exposure intensity. SEM photos of the SU-8 structures fabricated on glass wafers are illustrated in Figure 18a and Figure 19a, and the corresponding simulation results are shown in Figure 18a and Figure 19b. In practice, there are often multiple mask holes, and the spacing between the mask holes is an important parameter. Different mask hole spacing will affect the final development morphology of photoresist. 

Figure 18 shows the simulation and experimental results for linewidth/spacewidth of 30/30 μm. Through measurement and calculation, the top and bottom widths of the photoresist cylinder, shown in Figure 18a, were found to be 32.35 and 35.29 μm, respectively. Therefore, the top and bottom widths of developed SU-8 pillars were enlarged by 2.35 and 5.29 μm, respectively. Figure 19 shows the simulation and experimental results for linewidth/spacewidth of 30/50 μm. The top and bottom widths of the photoresist cylinder, shown in Figure 19a, were 31.78 and 33.08 μm, respectively. Therefore, the top and bottom widths of developed SU-8 pillars were enlarged by 1.78 and 3.08 μm, respectively. The width of the bottom of SU-8 glue is larger than that of the top. This is because the polymer is not very dense when the exposure dose at the bottom of SU-8 photoresist is relatively low. To compare the experimental and simulation results, the dimensions of simulation results were also measured according to the contour value of light intensity patterns. The top and bottom widths for linewidth/spacewidth of 30/30 μm, shown in Figure 18b, were 31.61 and 30.58 μm, respectively, while the top and bottom widths for linewidth/spacewidth of 30/50 μm, shown in Figure 19b, were 31.43 and 30.68 μm, respectively. A comparison between Figure 18a,b and Figure 19a,b showed that the simulation results exhibited a similar trend as the experimental results. However, the disparity between the experimental and simulation results in Figure 18 is notable. This was due to the decrease of mask hole spacing. The results revealed that the interaction between neighboring slits became more significant with diminution of the interval between two slits, and the diffraction effect was enhanced with a decrease in the interval between two slits, leading to the linewidth of the SU-8 structure with linewidth/spacewidth of 30/30 μm becoming a little larger than that with linewidth/spacewidth of 30/50 μm. 

SEM photos of the oblique SU-8 structures fabricated on glass wafers are illustrated in Figure 20a, and the corresponding simulation results are shown in Figure 20b. Figure 20a,b shows the intensity distribution of photoresist in the case of multiple mask holes with the incident angle of 30° and linewidth/spacewidth of 30/60 μm. From Figure 20a, we can see that the width of the top SU-8 was 31.75 μm, and the width of the bottom SU-8 was 33.40 μm. Therefore, the top and bottom widths of developed SU-8 pillars were enlarged by 1.75 and 3.40 μm, respectively. In particular, the variation of the bottom width was 1.75 μm. We also calculated the top and bottom widths in the simulation results. In comparison, the top width in the intensity pattern was approximately equal to that in the experimental result. Therefore we drew the conclusion that the simulation results were in good agreement with the experimental results. The measured structural inclined angles of developed SU-8 photoresists were 73.76° for the incident angle of 30°. Correlating Formula (12), the theoretical structural inclined angle α of developed structures was 72.58° for the incident angle of 30°, which was approximately the same as the measured value and thus demonstrated good agreement with the experimental results.

## 4. Conclusions

In order to simulate UV lithography and predict the profiles of SU-8 structures, a lithography simulation model based on the waveguide method and rigorous electromagnetic field theory is presented in this paper. According to this model, vertical and inclined UV exposure of SU-8 thick photoresist was successfully simulated, and the parameters significantly influencing light intensity distribution were studied. The presented model using the WG method works faster than other electromagnetic field methods and produces precise results. A series of experiments were designed to investigate the performance of the simulation model. By comparing simulation and experiments results, the method was successfully verified. This model is ideally suited for 2D lithography simulations for complex cases.

## Figures and Tables

**Figure 1 micromachines-11-00972-f001:**
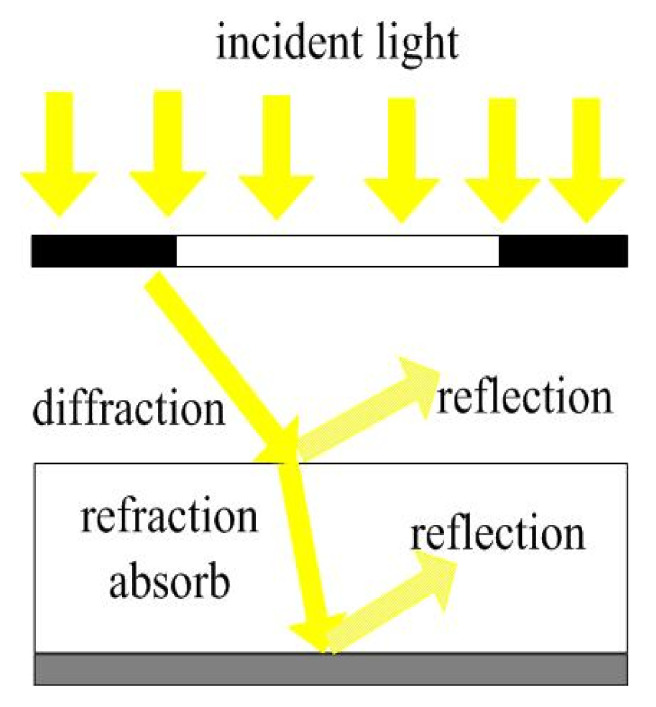
Light propagation process.

**Figure 2 micromachines-11-00972-f002:**
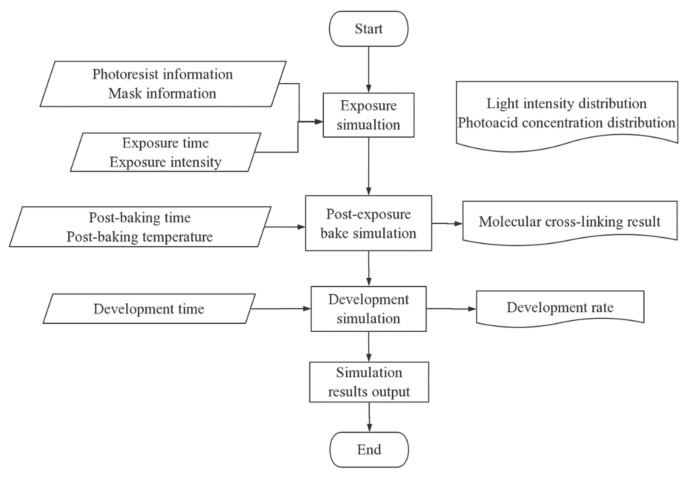
Flow chart of lithography simulation.

**Figure 3 micromachines-11-00972-f003:**
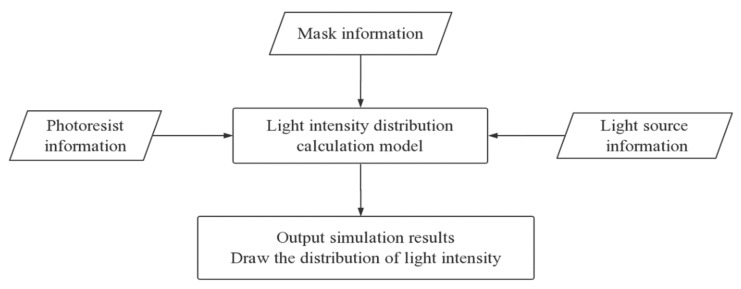
Model schematic.

**Figure 4 micromachines-11-00972-f004:**
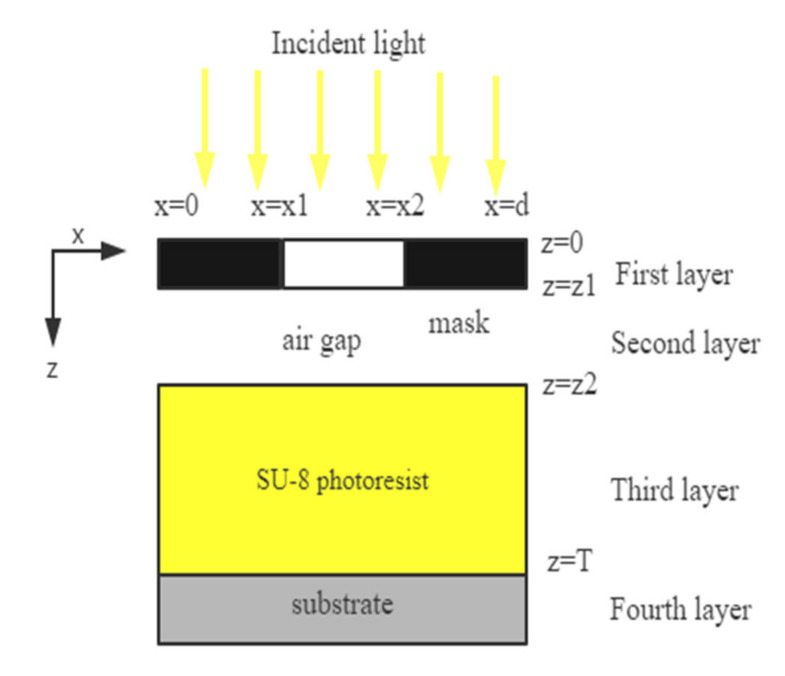
The two-dimensional schematic of the waveguide method.

**Figure 5 micromachines-11-00972-f005:**
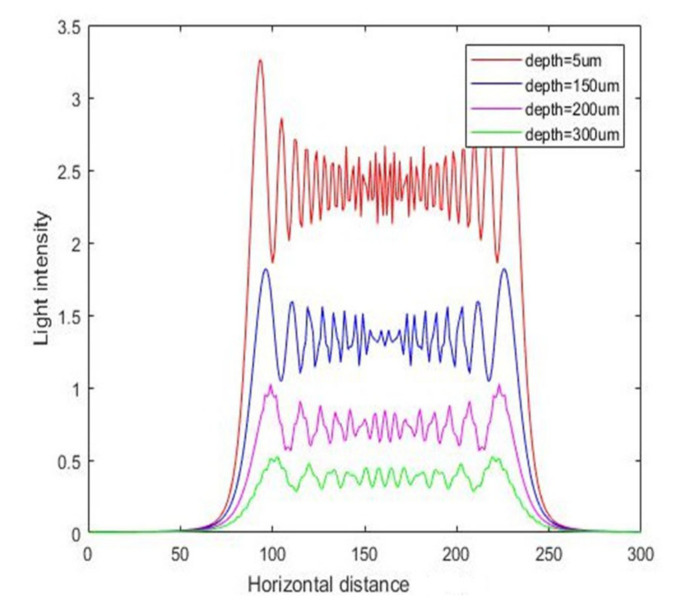
The intensity curve for vertical exposure with different depths.

**Figure 6 micromachines-11-00972-f006:**
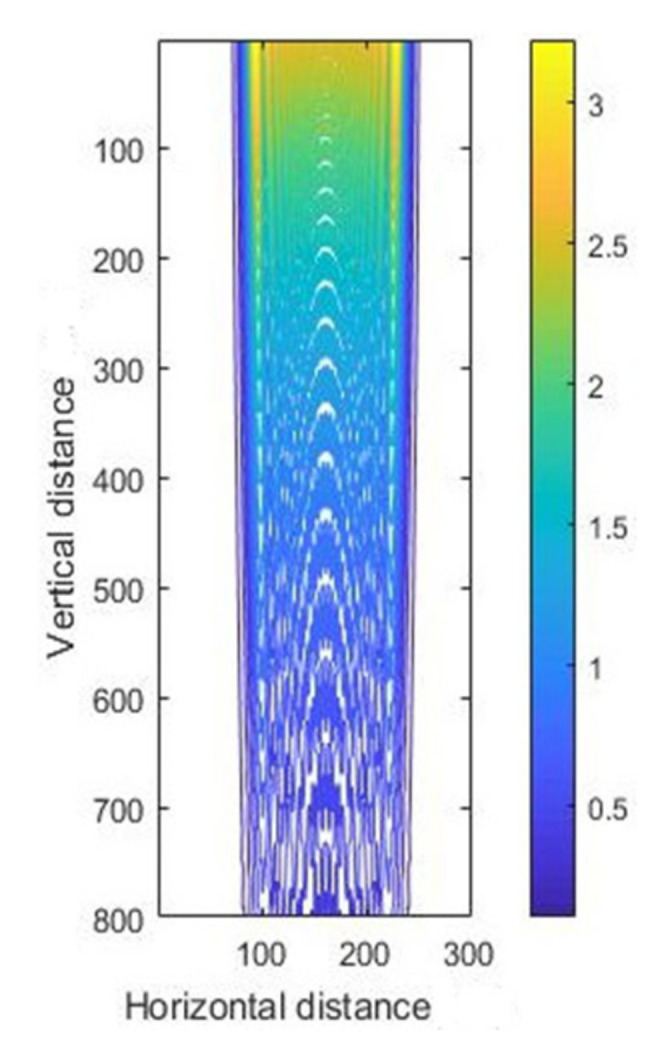
The contour map of intensity distribution for vertical exposure.

**Figure 7 micromachines-11-00972-f007:**
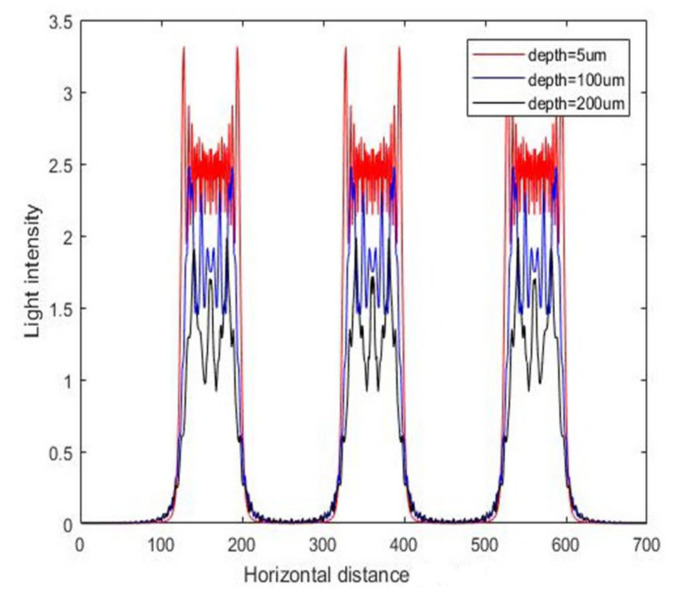
The intensity curve for vertical exposure with different depths.

**Figure 8 micromachines-11-00972-f008:**
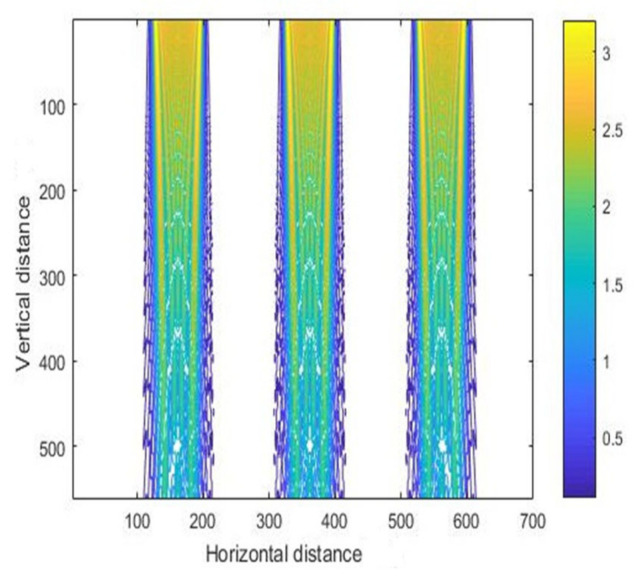
The contour map of intensity distribution for vertical exposure.

**Figure 9 micromachines-11-00972-f009:**
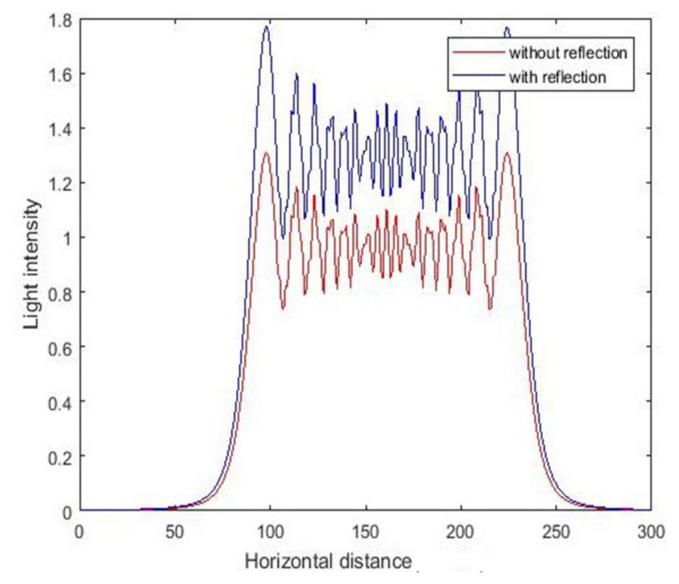
The intensity curve for vertical exposure at depth of 150 μm.

**Figure 10 micromachines-11-00972-f010:**
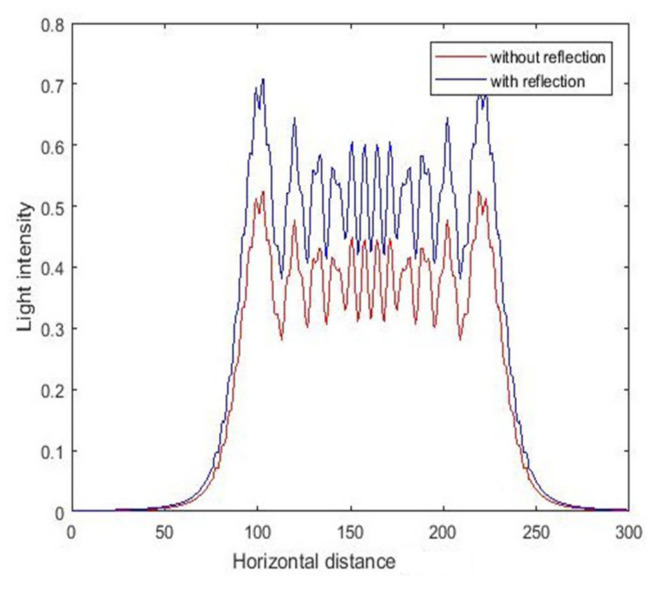
The intensity curve for vertical exposure at depth of 300 μm.

**Figure 11 micromachines-11-00972-f011:**
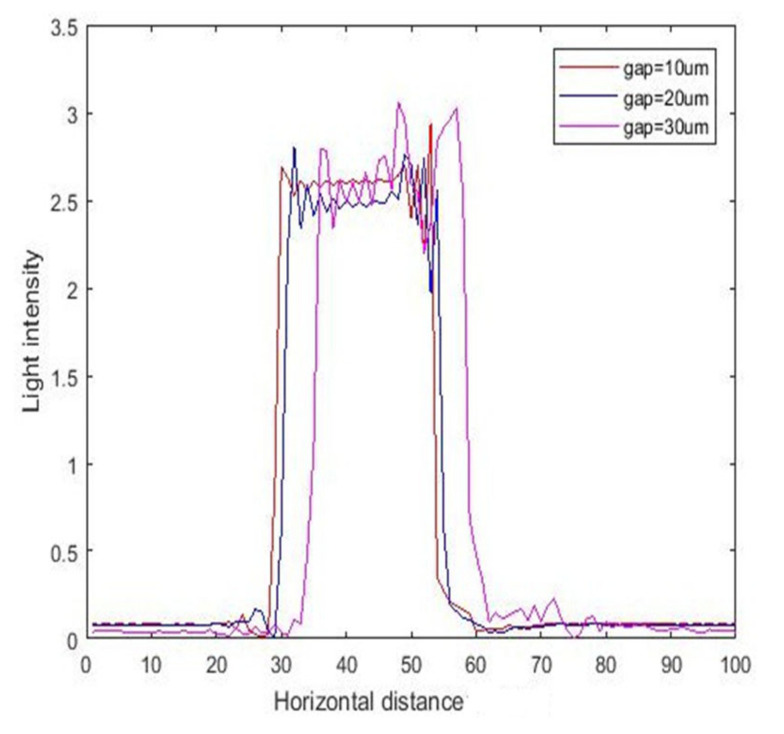
The intensity curve for inclined exposure with different air gaps (10, 20, and 30 μm) at 23.5° incident angle.

**Figure 12 micromachines-11-00972-f012:**
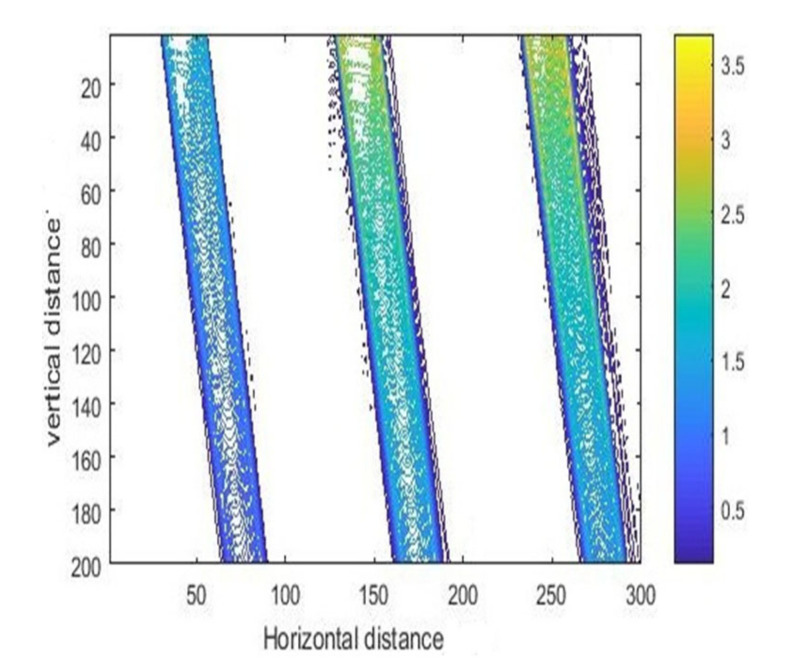
The contour map of intensity with different air gaps (10, 20, and 30 μm) at 23.5° incident angle.

**Figure 13 micromachines-11-00972-f013:**
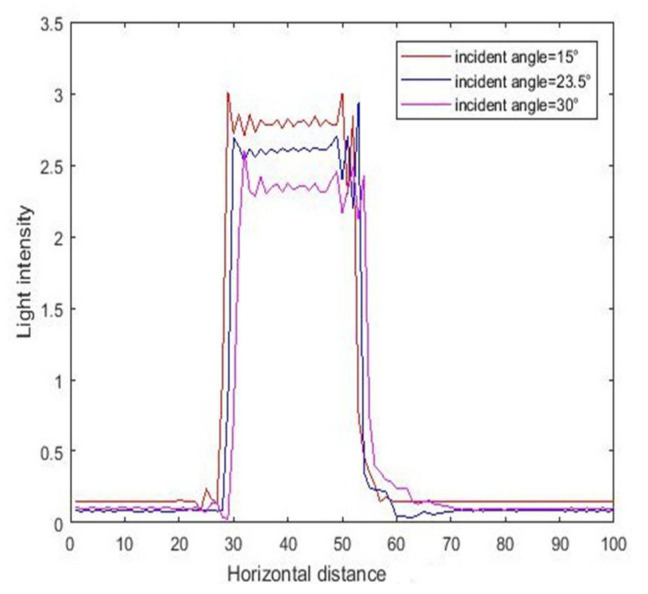
The intensity curve for inclined exposure with different incident angles in the X–Z cross section.

**Figure 14 micromachines-11-00972-f014:**
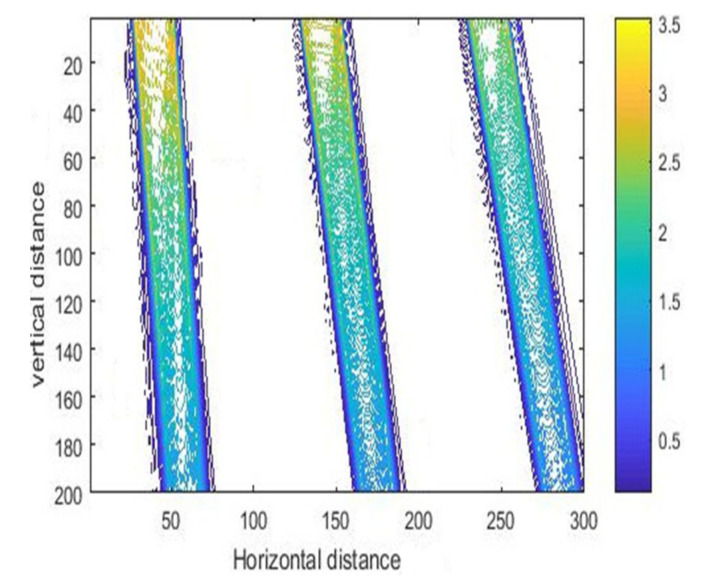
The contour map of intensity with different incident angles (15, 23.5, and 30°) with air gap of 10 μm.

**Figure 15 micromachines-11-00972-f015:**
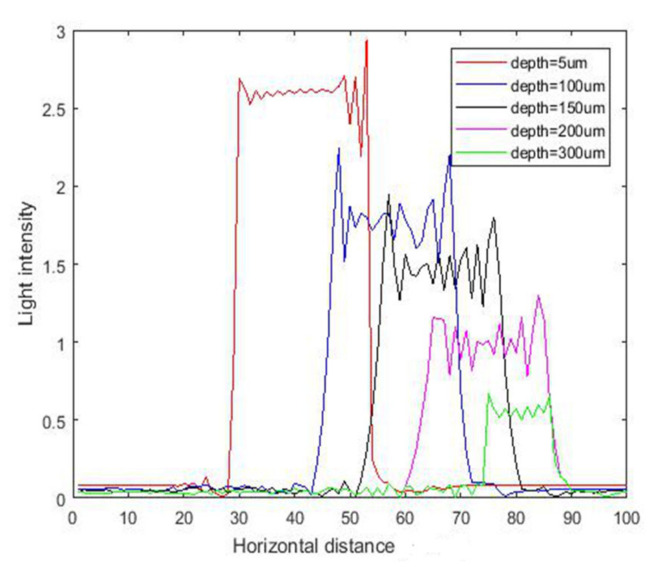
The intensity curve for inclined exposure with different depths at 23.5° incident angle.

**Figure 16 micromachines-11-00972-f016:**
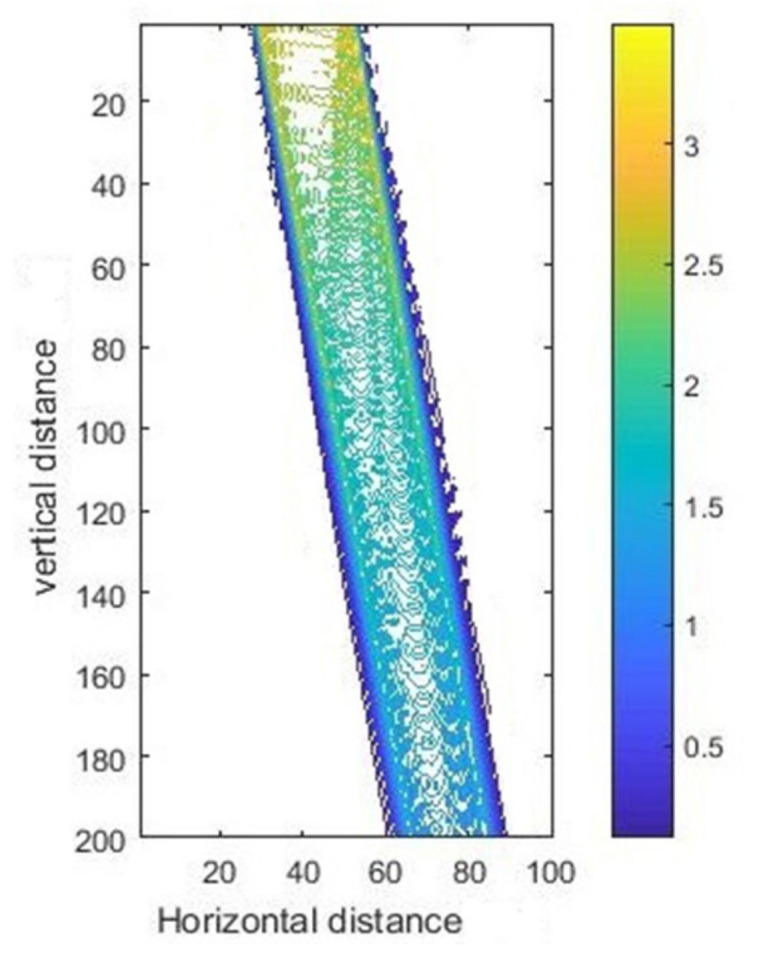
The contour map of intensity distribution for inclined exposure at incident angle 23.5°.

**Figure 17 micromachines-11-00972-f017:**
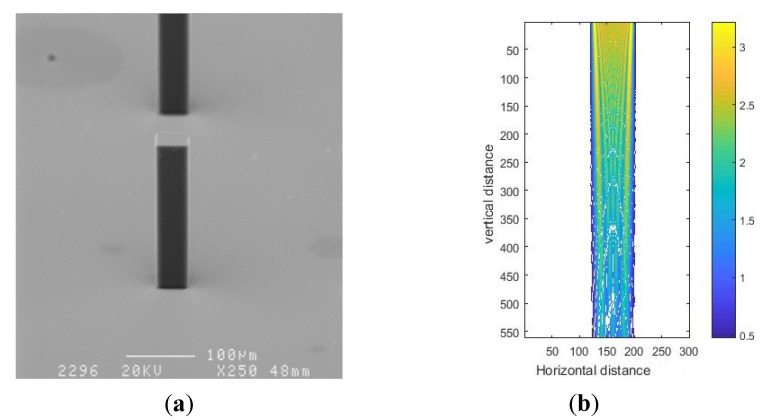
SEM photo of a SU-8 pillar (**a**) and simulation result (**b**) for mask hole size of 30 μm.

**Figure 18 micromachines-11-00972-f018:**
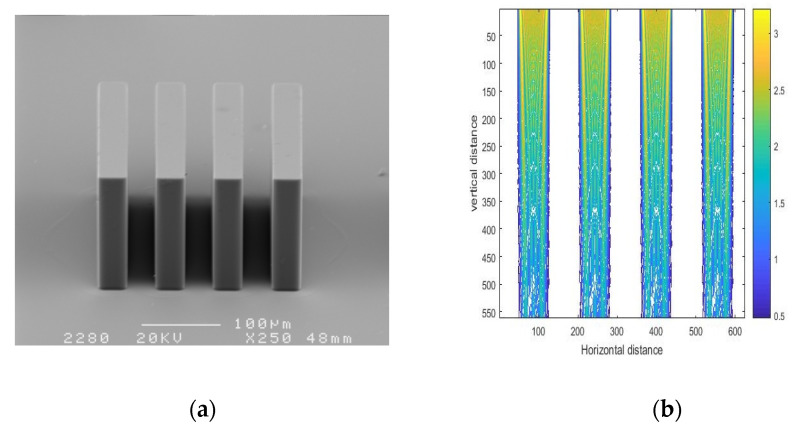
SEM photo of a SU-8 pillar (**a**) and simulation result (**b**) for linewidth/spacewidth of 30/30 μm.

**Figure 19 micromachines-11-00972-f019:**
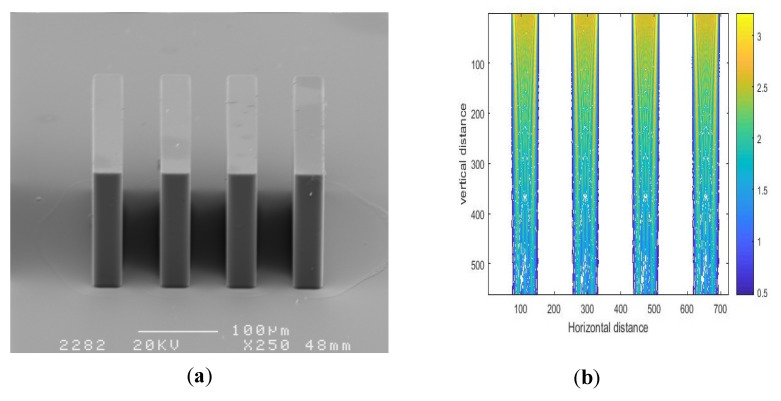
SEM photo of a SU-8 pillar (**a**) and simulation result (**b**) for linewidth/spacewidth of 30/50 μm.

**Figure 20 micromachines-11-00972-f020:**
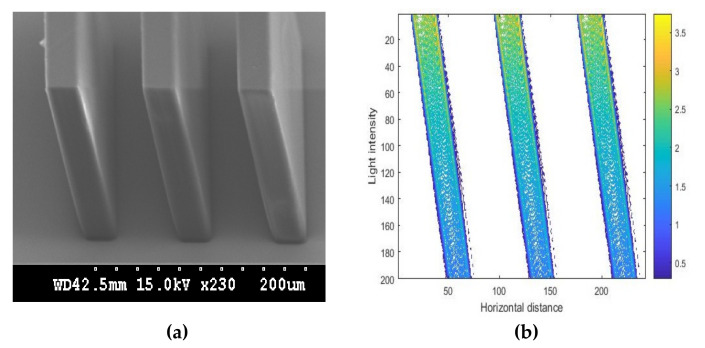
SEM photo of an oblique SU-8 pillar (**a**) and simulation result (**b**) for incident angle of 30° and linewidth/spacewidth of 30/60 μm.

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
