# Peer review of "A 2D Waveguide Method for Lithography Simulation of Thick SU-8 Photoresist"

_micromachines, 2020, doi:10.3390/mi11110972_

Round 1

Reviewer 1 Report

I agree with the authors’ premise that an understanding of the photolithographic exposure of SU-8 is important to be able to predict the resulting structuring.

To do so, light diffraction and reflection at interfaces need to be taken into account.

The critical dose of the SU-8 is a key factor, as it determines which parts of the exposed SU-8 will be polymerized to form a solid material which is chemically-resistant to the developer.

Concerning these two points above I suggest the authors revise their manuscript to take into account the findings of the following papers and include them in the citations:

Reflection at interfaces:

Gaudet, M., & Arscott, S. (2017). A user-friendly guide to the optimum ultraviolet photolithographic exposure and greyscale dose of SU-8 photoresist on common MEMS, microsystems, and microelectronics coatings and materials. Analytical Methods, 9(17), 2495-2504.

Critical dose:

Gaudet, M., Camart, J. C., Buchaillot, L., & Arscott, S. (2006). Variation of absorption coefficient and determination of critical dose of SU-8 at 365 nm. Applied Physics Letters, 88(2), 024107.

Relatively recent review articles concerning SU-8 are missing e;g.:

Arscott, S. (2014). SU-8 as a material for lab-on-a-chip-based mass spectrometry. Lab on a Chip, 14(19), 3668-3689.

(this article contains many references concerning SU-8)

The authors claim that the simulations are able to predict the photolithography of SU-8. As the simulations produce an intensity (dose) plot, given that there is a critical dose for SU-8 (see above) – the simulations should be able to predict specific SU-8 shapes e.g. a narrowing at the bottom of the pillars. However, this is not visible in the SEM images. It would thus be interesting to simulate reduced doses and see if the pillars are narrower at the bottom – this should correspond to a critical dose threshold (see above).

Author Response

Thank you for your good suggestions. Our manuscript was revised according to the your comments. Please see the attachment. Thank you very much!

Reviewer 2 Report

In this study, the authors introduce a lithography simulation of thick SU-8 photoresist environment using a 2D waveguide method. As the author pointed, light propagation will take different formations depending on variables such as distance between photomask and photoresist interface, photoresist depth, incident light angle, and pattern spacing on the photomask. Simulations that can rigorously predict these changes in advance are a powerful method to calculate the shape of the results before the production process. The results of the simulations performed by the authors and the fabricated structures in practice showed a very similar appearance, which assert that the simulation was precisely performed.

However, a specific figure of the actually fabricated structure is shown to reuse without copyright permission from previously published studies in other journals. Accordingly, it seems to claim that the light propagation simulation result is very similar after only performing the simulation without fabricating an actual structure.

Furthermore, different thresholds will be existed between light propagation and actual photohardening, but these are not mentioned in detail.

In addition, not only the scales of the simulations shown in Figures 7, 8, 15, 16, and 17 are different from the actually created structures but Figure 17(a) is mirrored. It seems that the manuscript is submitted after insufficient review by authors.

Therefore, I recommend the publication of this manuscript in micromachines, after the following major revisions.

(1) The Figure. 18(a) has been used a lot in previous studies. Isn't copyright permission required?

( Are there any other pictures used that have been published in other journals? )

( Journal of Micromechanics and Microengineering, 2008, 18.12: 125017.   Fig. 10(a) )

( IEEE transactions on semiconductor manufacturing, 2011, 24.2: 294-303.   Fig 8(b) )

( Microelectronic engineering, 2014, 123: 171-174.   Fig 5(b) )

(2) The authors only wrote about light intensity except for other details in this manuscript. Therefore, I recommend you to mention other specifications applied to the simulation and lithography processes.

(3) In lithography simulation, light propagation and photoresist hardening thresholds are needed to predict the structure actually fabricated. Is the hardening threshold of the photoresist the entire color marked part of the simulation contour map? If not, it would be better to add threshold lines to the contour map figure or add a new threshold figure.

(4) In Figure. 7 and 8, hole spacing and size are 40 and 70 μm, respectively. However, the scales between simulation and fabricated structure look different. Is there any reason why the scales are different in particular? (Figure. 15, 16, and 17 also too.)

(5) In line from 203 to 205, the authors mentioned that “The results reveal that the diffraction effect is enhanced with the decrease of the interval between two slits, leading to the line width of the SU-8 structure with line/space=30/30um becomes a little bit larger than the SU-8 structure with line/space=30/50um.”

Can the authors calculate the differences by the simulation? (How accurate is the simulation?)

(6) Depending on the light incident angles, the diffraction degrees would also have been different. What is the diffraction angle according to the angle of incidence? Does the result match the diffraction angle of the actually fabricated structure in Figure. 18(a)? In addition, what is the angle ‘a’ in the figure?

(7) Why is the picture in Figure. 17(a) mirrored? Is there any special reason?

(8) In line 232, 235 (Conflict of Interest) and reference 16, 20 are duplicated. It would be better to submit the revised version after careful revision by the authors.

(9) It is recommended to correct all ‘um’ to ‘μm’ of the manuscript.

Author Response

(The authors gave the same response as above.)

Reviewer 3 Report

This paper demonstrated the development of a simulation method to model the light intensity distribution over photolithography. The simulation results showed good agreement with the experiments and can guide future lithography process designs. Overall, a major revision is suggested for this manuscript before publication in Micromachines.

In the modeling, the input light is at a single wavelength of 365 nm, while the real light source has limited bandwidth. How will a broadband light source influence the light intensity profile? Especially, oscillatory field intensity distribution was observed in the results (Figure 5), which could be gone for a broadband light.

It is also not clear how the diffraction effect influences the light interactions between mask hole units at increased spacings. In general, what insights the model can provide to guide the lithography experiments to overcome this effect? The rounded corners are common features in photoresist posts after lithography. Is this noticeable in the experimental results?

Other comments are below:

  1. The axis labeling in Figures 6, 10, 12 and 14 should be consistent with Figure 5 in terms of the font size and style. Current fonts are stretched.
  2. The figure 17a is flipped and the texts are hard to read.
  3. There are too many figures in the main text. The authors can consider combining some of the figures or moving such as Figures 2, 3 to the supplementary information.
  4. How does the linewidth of SU8 posts varies at the top and bottom surfaces in Figures 16 and 17? These data will help better comparison with Figure 18.
  5. The authors should include the full name of MEMS in the manuscript.
  6. Will the modeling resources of simulation tools be available online for public access? It could be beneficial to the research community.

Author Response

(The authors gave the same response as above.)

Round 2

Reviewer 1 Report

Dear Authors,

Thank you for reponding to my remarks. I have a few points:

  1. The refererences need to put in the correct order.
  2. The journal proof reader can help with minor english e.g. on p4 [The] waveguide method.
  3. The accuracy of the thicknesses is odd 31.37 µm? 32.06 µm?
  4. I would also like to remind them of my comment regarding how nice it would be to show an SU-8 feature whose thickness varies - I thinbk that this would have been possible for a lower dose - although the Authors do mention reflection at the surface.

Good luck for your research!

Reviewer 3 Report

The authors have addressed the previous comments adequately. I would suggest publication of this manuscript.